# GABA_A_ Receptor/STEP61 Signaling Pathway May Be Involved in Emulsified Isoflurane Anesthesia in Rats

**DOI:** 10.3390/ijms21114078

**Published:** 2020-06-07

**Authors:** Xingkai Zhao, Guangjun Chang, Yan Cheng, Zhenlei Zhou

**Affiliations:** Department of Veterinary Clinical Science, College of Veterinary Medicine, Nanjing Agricultural University, Nanjing 210095, China; 2018107107@njau.edu.cn (X.Z.); changguangjun@njau.edu.cn (G.C.); 2017107113@njau.edu.cn (Y.C.)

**Keywords:** DNA methylation, emulsified isoflurane (EISO), epigenetic, general anesthesia, ion channel protein

## Abstract

(1) Background: Emulsified isoflurane (EISO) is a type of intravenous anesthetic. How emulsified isoflurane works in the brain is still unclear. The aim of this study was to explore whether epigenetic mechanisms affect anesthesia and to evaluate the anesthetic effects of emulsified isoflurane in rats. (2) Methods: Rats were randomly divided into four groups (*n* = 8/group): The tail vein was injected with normal saline 0.1 mL·kg^−1^·min^−1^ for the control (Con) group, with intralipid for the fat emulsion (FE) group, with EISO at 60 mg·kg^−1^·min^−1^ for the high-concentration (HD) group, and 45 mg·kg^−1^·min^−1^ for the low-concentration (LD) group. The consciousness state, motor function of limbs, and response to nociceptive stimulus were observed after drug administration. (3) Results: Using real-time polymerase chain reaction (PCR) to assess the promoter methylation of ion channel proteins in the cerebral cortex of rats anesthetized by EISO, we demonstrated that the change in the promoters’ methylation of the coding genes for gamma-aminobutyric acid A receptor α1 subunit (GABA_Aα1_), N-methyl-D-aspartate receptor subunit 1 (NMDAR1), and mu opioid receptor 1 (OPRM1) was accompanied by the change in messenger ribonucleic acid (mRNA) and protein expression by these genes. (4) Conclusion: These data suggest that the epigenetic factors’ modulation might offer a novel approach to explore the anesthetic mechanism of EISO.

## 1. Introduction

The earliest attempt at a non-inhalation method of administering volatile anesthetics was intravenous administration. Currently, most volatile anesthetics are purely liquid preparations. Clinical experiments have shown that volatile liquid anesthetics can be injected intravenously. However, this intravenous injection causes serious tissue and organ damage, and presents a variety of other complications [1]. Therefore, if volatile anesthetics are used to produce intravenous anesthetic effects, it is necessary to change the delivery method of the volatile liquid drugs. The development and wide application of fat emulsion products and the successful development of other low-water-soluble intravenous anesthetic emulsions, such as propofol, provide a new area for the use of intravenous anesthetics. Since the 1990s, anesthesiologists have studied the fat emulsion preparation of volatile anesthetics in depth, and successfully developed a stable isoflurane injection preparation, emulsified isoflurane (EISO), with a 30% intralipid emulsion as the solvent.

There are many advantages of delivering volatile anesthetics via intravenous injection compared to inhalational delivery. Intravenous injection of volatile anesthetics acts rapidly for the induction of anesthesia, by circumventing the anesthetic circuitry and the lung’s functional residual capacity. The injection of liquid volatile anesthetics is typically lethal to humans [2,3] and animals [4]. Adverse effects from the use of EISO were not observed in animals in previous studies [3,5]. EISO was used as an intravenous regional anesthesia in rat tails [6]. How the central nervous system was ultimately anesthetized by the intravenous injection of EISO, resulting in the loss of movement, pain, memory, and motor reflexes, is still unclear. DNA methylation is the transfer of a methyl group on S-adenosyl-L-methionine to a cytosine in a 5′-cytosine-phosphate-guanine-3′ (CpG) dinucleotide, with covalent linkages. DNA sequences containing a large number of CpG sites are often referred to as CpG islands, and there are many such CpG islands in mammalian genomic DNA, most of which are located in the promoter region. The methylated DNA can recruit methyl-CpG-bindingprotein-2 (MeCP2), which recognizes methylated cytosine in CpG dinucleotides, causing chromatin compression and gene silencing [7,8,9]. 

In recent years, many in vitro studies have shown that the mechanism of anesthesia may involve cell membranes, multiple receptors, ion channels, and neurotransmitters [10,11,12]. The N-methyl-D-aspartate (NMDA) receptor is an important excitatory amino acid receptor in the central nervous system. The gamma-aminobutyric acid A (GABA_A_) receptor is the most important inhibitory transmitter in the central nervous system, and the μ-opioid receptor (OPRM) is another important analgesic receptor in the central nervous system. The aim of our study was to explore whether epigenetic mechanisms are involved in regulating the activation of these receptors and their expression in the cerebral cortex, thereby affecting and regulating the expression of related ion channel proteins.

## 2. Results

### 2.1. Changes in Physiological Parameters after Emulsified Isoflurane Administration

In general, EISO injection significantly reduced the body temperature and reduced the respiratory rate (Appendix A). In addition, there was no significant difference in the body temperature or respiratory rate between the control (Con) group and fat emulsion (FE) group, or the low-concentration (LD) group and high-concentration (HD) group. The general monitoring results for anesthesia are shown in Table 1. The palpebral reflex (PR) results in the Con and FE groups were lower than those in the LD and HD groups. The corneal reflex (CR) results in the Con and FE groups were significantly lower than those in the LD and HD groups, and the CR result in the LD group was significantly lower than that in the HD group. Regarding the perianal reflex (PeR), the levels of PeR in the Con and FE groups were lower than those in the LD and HD groups. Regarding the reflective effects scores, as shown in Table 2, the sedation scores (SS), analgesic scores (AS), muscle relaxation scores (MRS), posture scores (PS), auditory response scores (AR), and total scores (TS) in the HD and LD groups were significantly higher than those in the Con and FE groups. In addition, the AS in the HD group was higher than that in the LD group. In general, the total score was more than 11 points, which indicates that a good anesthetic effect was achieved. 

### 2.2. Messenger RNA (mRNA) Expressions in the Tissue of the Parietal Lobe of the Cerebral Cortex 

Regarding the expression of the receptor-encoding gene, the mRNA expressions of GABA_A_α1 and Oprm1 from the cerebral cortex in the HD and LD groups were higher than those in the Con and FE groups, while N-methyl-D-aspartate receptor subunit 1 (NMDAR1) and N-methyl-D-aspartate receptor subunit 2B (NR2B) showed the opposite (Figure 1A). There was no significant difference between the FE and Con groups.

Regarding transcription factors, such as extracellular signal-regulated kinase 1/2(ERK1/2), c-Jun N terminal kinase (JNK), protein 38 (P38), and nuclear factor kappa-B (NF-κB), the mRNA expressions of the factors in the HD group were significantly higher than those in the Con group (Figure 1B). The expressions of ERK and p38 mRNA in the LD group were significantly higher than those in the Con group. In addition, the expressions of JNK and NF-κB mRNA in the HD group were significantly higher than those in the FE groups. There was no significant difference in the expressions of those factors between the FE and Con groups. 

As for the neurological function factors, the expressions of striatal-enriched protein tyrosine phosphatase 61 (STEP61) and Notch mRNA in the LD and HD groups were significantly higher than those in the Con group, and the tyrosine kinase FYN expression was the opposite (Figure 1C). The expression of Notch mRNA in the FE group was significantly higher than that in the Con group. The expression level of Notch in the HD group was significantly higher than that in the FE group.

### 2.3. Protein Results of the Candidate Receptor

Our results showed that the administration of EISO remarkably upregulated the protein expressions of GABA_A_α1 and OPRM1 in the HD and LD groups compared to the Con and FE groups. However, the NMDAR1 expression was significantly downregulated in the HD and LD groups compared to the Con and FE groups (Figure 2). There was no expression difference in those protein expressions between the HD and LD groups. This indicates that EISO induced anesthesia by regulating the expression of ligand-gated channel-related proteins. We used immunofluorescence to observe the location of the target proteins. ESIO administration enhanced the expressions of GABA_A_α1 and OPRM1, but weakened the expression of NMDA in the HD and LD groups compared to the Con and FE groups (Figure 3).

### 2.4. Methylation Analysis in the Promoter Region of Candidate Genes

The promoter region of the GABA_A_α1, Oprm1, and NMDAR1 genes was found by comparing the mRNA sequence with the genome sequence. Then, according to the analysis results of the transcription factor binding sites in the promoter region by the Matinspector in the Genomatix online tool, the core promoter structure of GABA_A_α1, Oprm1, and NMDAR1 was determined (Figure 4A). The promoter sequences of those genes were found in National Center for Biotechnology Information (NCBI). Subsequently, the potential binding sites of the relevant transcription factors and DNA methylation sites in the promoter regions of these genes were analyzed to confirm the core promoter sequences determined by methylation. The methylation rates of the promoter regions of GABA_A_α1 and OPRM1 in the HD and LD groups were lower than those in the Con and FE groups, whereas, NMDAR1 demonstrated the opposite (Figure 4B).

## 3. Discussion

The circulation and respiratory system are two systems that are affected by anesthetic drugs [13]. The study of intravenous emulsified isoflurane indicated that there were no adverse reactions found except for caudal local skin ulcers. [14] In our study, rats were injected with 60 mg·kg^−1^·min^−1^ and 45 mg·kg^−1^·min^−1^ emulsified isoflurane through the tail vein. The body temperature and respiratory frequency showed a downward trend after injection. These indicated that EISO had a significant effect on the rat circulation and respiratory system during anesthesia. In the detection of the eyelid reflex, there was no observed phenomenon of dodging, closed eyes, or blinking, which proved that the rats were in the clinical anesthesia stage period. The rats had no eyeball contractions, tremors, or blinking in the corneal reflex test and no obvious anal contraction in the anal reflex test. 

The results proved that the rats were in a safe and appropriate state of anesthesia. In the monitoring of the analgesic effects, the rats in the acupuncture and clamp detection sites did not have any pain reaction, or sedative or muscle relaxation effects. During the period, this state met the requirements for clinical surgery. Liu found that the minimum alveolar concentration (MAC) of EISO by intravenous injection in beagle dogs was lower than that during isoflurane inhalation anesthesia [15]. We speculate that the difference in the concentration gradient of anesthetics in the blood and alveoli as caused by different routes of administration is related to the balance time, not the change in anesthetic efficacy. 

Regarding the tactile pain-related perceptions in the parietal cortex, we selected this tissue to study the anesthetic mechanism of emulsified isoflurane at the molecular level. Recent studies proved that ion channel modulation is the basis for general anesthesia [16]. The GABA_A_ receptor is an ion channel sensitive to volatile anesthetics with clinically available concentrations, and this receptor belongs to the superfamily of cysteine ring neurotransmitter receptors. It is also the most abundant inhibitory neurotransmitter receptor in the brain [17]. Studies demonstrated that isoflurane has a direct stereoselectivity for the GABA_A_ receptor, which indicates that isoflurane may directly combine with a subunit on the GABA_A_ receptor to produce a physiological effect [18]. Previous studies indicated that two different α subunits or two different β subunits were present in at least some GABA_A_ receptors and that 98% of α1 receptors contained either γ1, γ2, γ3, or δ subunits [19]; hence, GABA_A_α1 was the target of our study. 

We also found that both GABA_A_α1 protein expression and mRNA expression increased significantly after EISO injection. STEP61, a tyrosine phosphatase specifically expressed in neurons, was also a downstream molecule of the GABA_A_ receptor and increased significantly after EISO injection. The increased expression of STEP61 reduced the molecular binding between ERK and FYN. Many studies have shown that ERK also plays a key regulatory role for NMDA receptors, and the increased expression of ERK can promote the activation of NMDA receptor phosphorylation [20]. However, in our study, we found that ERK expression increased with the decrease in NR2B expression. We speculated that the intralipid influenced the expression of ERK. NR1 and NR2B were both important substrates of STEP61, and FYN-mediated phosphorylation of NR2B-Tyrl472 could promote NR2B aggregation to synapses, which was negatively regulated by STEP61 [21]. This indicated that EISO further affects the expression of NR1 and NR2B subunits by activating the GABA_A_/STEP pathway, thereby downregulating the expression of the NMDA receptors and participating in the transmission and regulation of pain information in the cerebral cortex.

To further explore the anesthetic mechanism of EISO, we explored DNA methylation detection, which is an epigenetic event and plays a key role in the regulation of gene expression [22]. We found that the methylation degree of the promoter regions of the coding genes of GABA_Aα1_, NMDAR1, and OPRM1 subunits changed significantly after EISO injection, and this methylation regulated the expression of these three subunits. However, it was not clear how EISO regulated the methylation of the GABA_Aα1_, NMDAR1, and OPRM1 promoters. The methylation of the promoters of these coding genes altered the degree of DNA compression, thereby affecting the binding rate of transcription factors and altering the expression of the proteins encoded by these genes. We also found that NF-κB and cAMP-response element binding protein (CREB), as downstream molecules of P38, were important transcription factors of these three target genes, which were also significantly upregulated in the experiment. However, the expression of NMDAR1 was decreased after the injection of EISO. This indicated that DNA methylation altered gene expression more than transcription factor expression altered gene expression in our study. Our protein experiments also confirmed this hypothesis.

A previous study showed that isoflurane produced both experimental neuroprotective and neurotoxic effects on the developing brain, depending on the duration and level of exposure, and EISO, as an intravenous anesthetic, may have protective and injurious effects on rats [23]. In both in vivo and in vitro ischemia models, mitogen-activated protein kinase (MAPK) was demonstrated to play a crucial role in regulating brain cell death and survival after ischemia [24]. In previous studies, OPRM was found to be directly regulated by MAPK. In our study, we found that EISO affected the expression of the transcription factors CREB and NF-κB through the MAPK signaling pathway, further affecting the expression of three target genes [25,26,27]. In future studies, we will continue to investigate which transcription factors play key roles while observing the degree of chromatin loosening after the transcription factor binds to the coding gene to better explore the epigenetic mechanisms by which EISO causes anesthesia.

In conclusion, we found that after the anesthesia of rats with EISO, the expression of related factors in the GABA_A_ receptor/STEP61 signaling pathway changed, the expression of the GABA_A_ receptor increased, and the expression of the NMDA receptor decreased, accompanied by a change in the promoter methylation. The MAPK signaling pathway might also be involved in regulation. More work is needed in the future to determine the sequence and causality of downstream factors of the GABA_A_ receptor/STEP61 signaling pathway.

## 4. Methods

Experimental Sprague Dawley (SD) rats were housed in individual cages in the Centre of Experimental Animals at Nanjing Agricultural University (Nanjing, China) in accordance with the guidelines of Animal Ethics Committee at Nanjing Agricultural University in compliance with the Regulations for the Administration of Affairs Concerning Experimental Animals (The State Science and Technology Commission of People’s Republic of China, 1988). All experimental protocols were approved by the animal Care and Use Committee of Nanjing Agricultural University, 1999(#NJAU-Rattus-2018061305, approved on 13 June 2018).

### 4.1. Animals

We used 32 male SD rats, 220–230 g, 2–3 months old for this study. All rats were housed separately under standard laboratory conditions with a 12:12 h light/dark cycle, 22 °C, and 50% humidity. The rats had free access to water and standard mouse chow. Prior to the investigations, the rats were allowed to habituate to their new surroundings for one week after having been transferred from the breeder. Our study was approved by the Nanjing Agricultural University animal ethics committee. The following experiments were conducted.

### 4.2. Drugs

Isoflurane emulsions of eight percent by volume were prepared by adding pure isoflurane to 30% intralipid at 20 °C. The solution was then vortexed at 50 Hz every five minutes for one minute, lasting two hours, stored at 4 °C between injections, and warmed to 20 °C.

### 4.3. Experimental Design

The rats were fasted for 12 h before the experiment but free to drink water. The rats were randomly separated into four groups (*n* = 8): Con, FE, HD, and LD. The caudal vein was injected with saline solution (0.1 mL·kg^−1^·min^−1^), 30% fat intralipid (0.1 mL·kg^−1^·min^−1^), or 8% EISO (60 or 45 mg·kg^−1^·min^−1^), with a constant flow syringe pump to maintain anesthesia for 25 min, and the anesthetic conditions of the rats were assessed. In the experiment, the doses of EISO used for maintaining anesthesia and for inducing anesthesia were the same. The HD group used EISO at 60 mg·kg^−1^·min^−1^, the LD group was at 45 mg·kg^−1^·min^−1^, and the process of anesthesia was relatively stable. Tissue samples from the parietal lobe of the cerebral cortex were taken from rats after decapitation and preserved in liquid nitrogen.

### 4.4. Physiological Indicator Test

After the start of administration, the body temperature and respiratory rate of the rats were measured in five-minute intervals. 

### 4.5. Monitoring of Biological Reflection

Observing and monitoring reflexes, such as the corneal reflex (CR), and the anal reflex bluntness, and disappearance, have been used as one of the main methods to evaluate the depth of anesthesia. The corneal reflex mainly stimulates the cornea through a hard lane to observe whether there are eye contractions, tremors, eye movements, and other phenomena. The anal reflex (AR) refers to the contraction of the anal sphincter when the anus is suddenly stimulated. The detailed scoring criteria is described in Appendix A. 

### 4.6. Monitoring of Braking and Analgesic Effects

The scoring criteria used to investigate the brake and analgesic effects included sedation, analgesia, muscle relaxation, posture, and hearing sensitivity. Sedation was scored based on the position of the eyeball and the sensitivity to the external environment. Analgesia was judged by the animal reaction to clamping the animal’s toes with a hemostat. Muscle relaxation was judged by the tension of the masseter muscle and leg muscles. The posture was scored by observing the lying posture of the small rat. Hearing induction was evaluated by clapping hands near the ear of the rat and observing the reaction. The detailed scoring criteria was described in previous studies and are shown in Appendix A. Evaluations of the brake and analgesic effects were made in five-minute intervals after the beginning of drug injection and the resulting scores were recorded.

### 4.7. Quantitative Real-Time Polymerase Chain Reaction (PCR)

The total ribonucleic acid (RNA) was extracted from the tissue with the RNA iso Plus™ (cat. 9108, Takara, Dalian, Liaoning Province, China) reagent according to the manufacturer’s instructions. The complementary deoxyribonucleic acid (cDNA) was prepared using a reverse transcription kit (cat. RR036A, Takara, Dalian, Liaoning Province, China) according to the manufacturer’s instruction. The cDNA was subjected to quantitative PCR amplification for the transcription of candidate genes. The primers are shown in Appendix A. Each RNA sample was processed in two copies using an amount of DNA equivalent to the total RNA, starting at 250 ng. The target sequences of the candidate genes were amplified with specific primer pairs using the SYBR Premix Ex Taq Kit (cat. DRR420A, Takara, Dalian, Liaoning Province, China). Amplifying PCR and monitoring of the fluorescent emission in real time were performed using the ABI Prism 7300 Sequence Detection System (Applied Biosystems, Inc., Foster City, CA, USA).

### 4.8. Genomic DNA Methylation Status Monitoring

We obtained the genomic DNA of the brain cortex using a DNA extraction kit (KG203, TIANGEN BIOTECH Co., Ltd., Shanghai, China). The target fragment was subjected to restriction analysis using DNAMAN software (Lynnon Biosoft, San Ramon, CA, USA), and the genomic DNA was pre-cut by restriction enzymes that did not cleave the target fragment. This study used E*co*RI (cat. 1040S, Takara, Dalian, Liaoning Province, China) to pre-cut the genomic DNA. We used M*sp*I and H*pa*II (cat. 1150A, Takara, Dalian, Liaoning Province, China) to process the mass of each sample equalized after digestion, and each sample was made with three parallels, M*sp*I, H*pa*II, and a blank control tube. Quantification of the absolute copy number of the genomic DNA was performed after enzyme treatment with RT-PCR and the calculation of the methylation of sample’s promoter. The primers used for methylation detection are listed in Appendix A.

### 4.9. Western Blotting

The proteins in the tissues were separated by sodium dodecyl sulfate-polyacrylamide gel electrophoresis using a 10% acrylamide gel and transferred to a polyvinylidene fluoride membrane by electroelution. The membranes were blocked with 5% non-fat milk powder in 0.01 M PBS (pH 7.4) and 0.1% Tween-20 at room temperature for 2 h. Substitute blots for members were incubated with primary antibodies against GABA_A_α1 (1:200; sc-7348, Santa Cruz, Dallas, TX, USA), NMDAR1 (1:200; sc-1468, Santa Cruz, Dallas, TX, USA), Oprm1 (1:200; sc-247994, Santa Cruz, Dallas, TX, USA), and glyceraldehyde-phosphate dehydrogenase (GAPDH) (ab8245, Abcam, Cambridge, MA, USA) overnight at 4 °C by gentle shaking, followed by the addition of the appropriate horseradish peroxidase-conjugated secondary antibodies (1:2000; A0208, Beyotime Biotechnology, Shanghai, China) and incubation at room temperature for 4 h. The expression levels of the proteins were quantitatively analyzed through comparison with GAPDH as an internal control. We used the Bio-Rad Gel Doc 2000 system analysis software (Bio-Rad, Hercules, CA, USA) to conduct the grey level of the bands.

### 4.10. Immunofluorescence

Paraffin sections were made of the cerebral cortex. The following primary antibodies were used for dual fluorescent labeling, GABAAα1, NMDAR1, and OPRM1. Subsequently, we observed the sections under a fluorescence microscope with different laser wavelengths.

### 4.11. Statistical Analysis

The data from the assessments of biological reflections and braking and analgesic effects were analyzed as repeated measures using the MIXED procedures of SAS (SAS version 9.4, SAS Institute Inc., San Diego, CA, USA). The effects of fat emulsions and emulsified isoflurane were considered fixed and the effect of the scoring time was analyzed as a repeated measure. The effect of the experimental rates was considered a random effect. In addition, the data of mRNA and protein expression, and the percentage of promoter methylation, were analyzed using the ANOVA package of SAS. All data were represented as the mean ± standard error (SE). The effects were considered significant at *p* < 0.05 and highly significant at *p* < 0.01. Trends were discussed at *p* < 0.10.

## 5. Conclusions

Emulsified isoflurane is a good anesthetic for intravenous injection. After producing anesthesia in rats with EISO, the expression of related factors in the GABA_A_ receptor/STEP61 signaling pathway changed, the expression of the GABA_A_ receptor increased, and the expression of the NMDA receptor decreased, accompanied by a change in the promoter methylation. The MAPK signaling pathway may also be involved in regulation. More work is still needed in the future to determine the relationship between the downstream molecules of this signaling pathway and to explore the relevant mechanisms by which EISO affects promoter methylation.

## Figures and Tables

**Figure 1 ijms-21-04078-f001:**
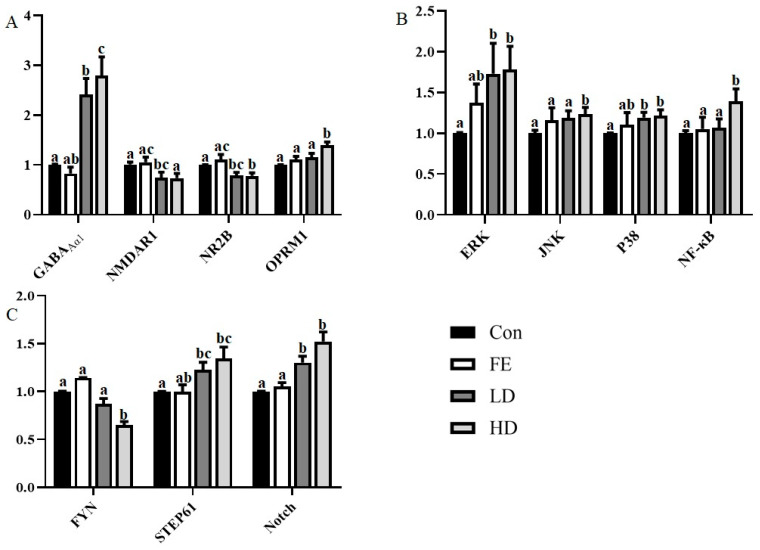
The messenger RNA (mRNA) expression of related genes encoding ion channels and receptor related proteins (**A**), coding transcription factor-related genes (**B**), and other cytokines (**C**). Different colors are used to indicate the different processing methods. β-actin was used as the reference gene for normalization, and the difference was calculated with STATISTICAL ANALYSIS SYSTEM (SAS). Superscripts with different letters indicate significant differences (*p* < 0.05).

**Figure 2 ijms-21-04078-f002:**
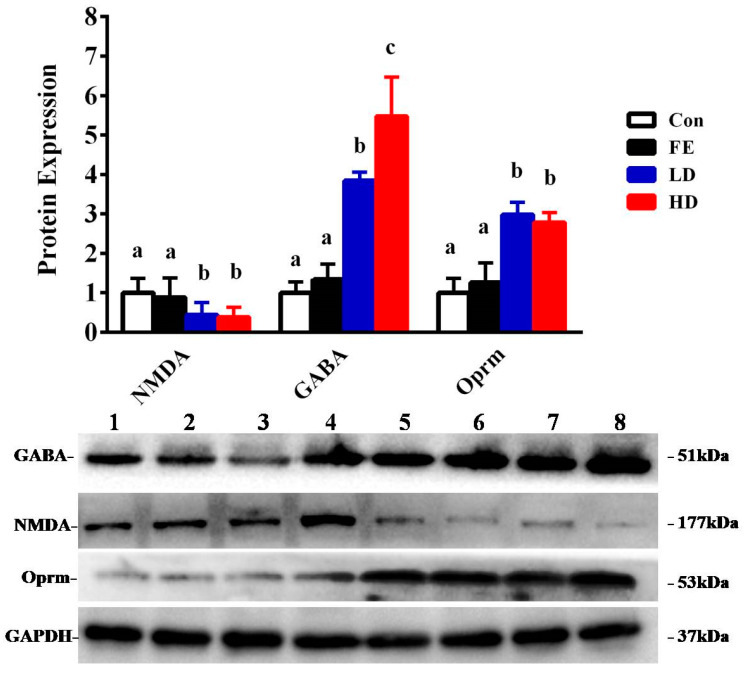
The determination of protein expression in the parietal lobe of Sprague Dawley (SD) rats injected with different drugs. The control (Con) group is indicated by a blank bar, the fat emulsion (FE) group is indicated by a black bar, the low-concentration (LD) group is indicated by a blue bar, and the high-concentration (HD) group is indicated by a red bar. Compared to the protein expression of the Con group and FE group, the LD group and HD group show significant changes. Superscripts with different letters indicate significant differences (*p* < 0.05). The left-to-right bands of each protein represent the corresponding protein expression levels in the parietal cortex samples of SD rats injected with different drugs and concentrations.

**Figure 3 ijms-21-04078-f003:**
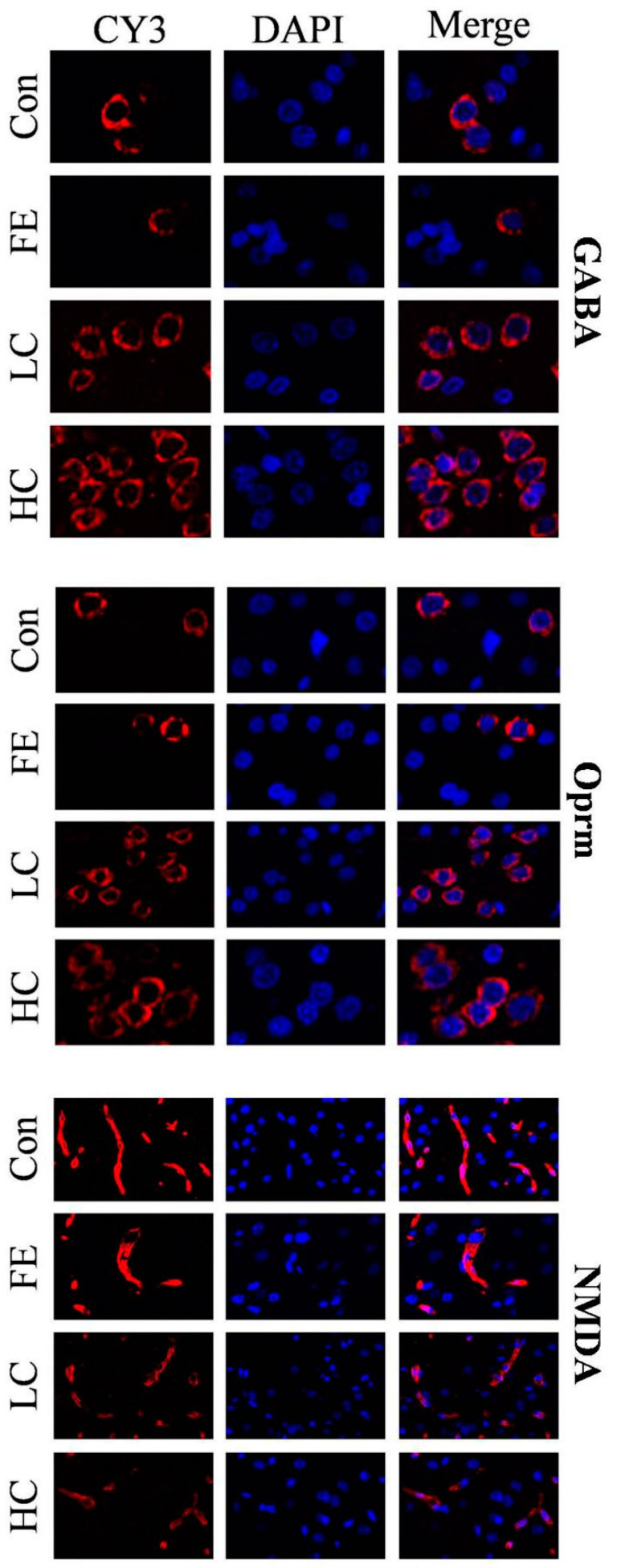
After treatments with different treatment groups, the expression levels and localization of the target protein in the parietal tissue of the cerebral cortex of SD rats. CY3 fluorescence detected gamma-aminobutyric acid A receptor 1 (GABA_A_1), N-methyl-D-aspartate receptor subunit 1 (NMDAR1), and μ-opioid receptor 1 (OPRM1) (red), and DAPI detected cell nuclei (blue). These data represent three independent experiments.

**Figure 4 ijms-21-04078-f004:**
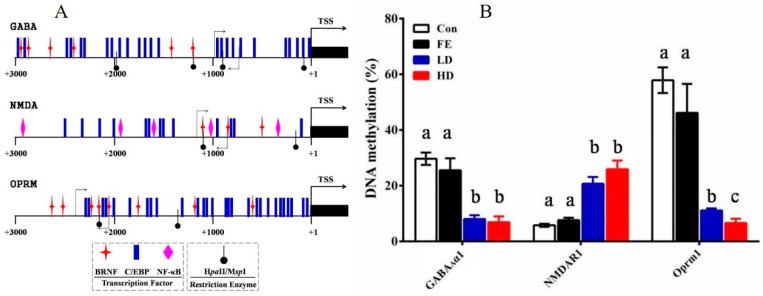
DNA methylation detection in the promoter area of GABA_A_α1, NMDAR1, and OPRM1 in the parietal lobe of cerebral cortex samples from rats injected with different medicine. The structure of the primary transcription factor binding sites and DNA methylation positions in the promoter area of the candidate genes. Different symbols are used to indicate the positions of the transcriptional factors and restriction enzymes. The primer pairs used for the methylation assay are denoted by gray arrows (**A**). The detection results of the methylation rates of the GABA_A_α1, NMDAR1, and OPRM1 promoter regions. Different colors are used to indicate the different processing methods. The difference was calculated with SAS. The results are expressed as the mean ± SE. Superscripts with different letters indicate significant differences (*p* < 0.05) (**B**).

**Table 1 ijms-21-04078-t001:** The general monitoring results of Sprague Dawley (SD) rats anesthetized by emulsified isoflurane injection.

Item	Con	FE	LD	HD	SE	*p* Value		
						Treated	Time	Tr × T
PR	1.00 a	1.00 a	2.85 b	2.93 b	0.04	<0.01	<0.01	<0.01
CR	1.00 a	1.00 a	2.88 b	2.95 c	0.03	<0.01	<0.01	<0.01
PeR	1.00 a	1.00 a	2.93 b	2.93 b	0.04	<0.01	<0.01	<0.01

Con, Control; FE, Fat Emulsion; LD, Low Dose; HD, High Dose; PR, Palpebral reflex; CR, Corneal reflex; PeR, Perianal reflex; Tr × T, Treated × Time; SE, standard error; Means in the same row with different Roman letters (a, b, and c) represent significant differences.

**Table 2 ijms-21-04078-t002:** Scores of the reflective effects of emulsified isoflurane administrated by intravenous infusion in SD rats.

Item	Con	FE	LD	HD	SE	*p* Value		
						Treated	Time	Tr × T
SS	0.00 a	0.00 a	2.80 b	2.80 b	0.05	<0.01	<0.01	<0.01
AS	0.00 a	0.00 a	2.90 b	2.93 c	0.03	<0.01	<0.01	<0.01
MRS	0.00 a	0.00 a	2.93 b	2.93 b	0.03	<0.01	<0.01	<0.01
PS	0.00 a	0.00 a	3.00 b	3.00 b	0.03	<0.01	<0.01	<0.01
AR	0.00 a	0.00 a	2.93 b	2.93 b	0.03	<0.01	<0.01	<0.01
TS	0.00 a	0.00 a	14.55 b	14.58 b	0.08	<0.01	<0.01	<0.01

Con, Control; FE, Fat Emulsion; LD, Low Dose; HD, High Dose; PR, Palpebral reflex; CR, Corneal reflex; PeR, Perianal reflex; Tr × T, Treated × Time; SE, standard error; Means in the same row with different Roman letters (a and b) represent significant differences.

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
