# Peer review of "GABAA Receptor/STEP61 Signaling Pathway May Be Involved in Emulsified Isoflurane Anesthesia in Rats"

_ijms, 2020, doi:10.3390/ijms21114078_

Round 1

Reviewer 1 Report

The manuscript is interesting and the experiments appear to be well thought out leading to important conclusions.  The English language is very poor.  The manuscript will require a careful edit by an English language expert before being considered for publication.  I have listed some of the changes that must be made:

Line 15 change to"...clear how emulsified..."

Line 16.  affect (not affecting

Line 26 "...(OPRMI) was (omit comma and add was)

Line 36.  However, this (lower case T)

Line 37 Therefore if volatile anesthetics are used to (substitute are used for want)

Line 40 such as propofol (omit parenthesis add such as) would provide a new area for the use of intravenous anesthetics.

Line 47.  Substitute "with" with "by"

Line 48.  The injection of liquid volatile anesthetics is usually lethal to humans2,3 and animals4.

Line 63. transmitter

Line 63. is another

Line 73. "or the"  substituted for "as well as"

Line 91 Regarding transcription factors

Line 96 There was no

Line 118.  References are required for have been assessed in rats (references)

Line 119 Omit Hence and start with The promoter

Line 166 omit more

Line 173 closed eyes or blinking omit comma

Line 173,  Period after stage

Line 174.  Start with Capital T The rats had no eyeball contraction, tremor or blinking omit (comma)

Line 176. anesthesia state. 

Line 176.  In the analgesic...

Line 178.  it met the requirements for  clinical surgery. (omit can meet and

routine)

Line 180.  anesthesia. We speculate

Line 205 we explored (omit through)

Line 214.  omit our idea.  Substitute this hypothesis

Line 223 we will continue to

Line 239.  We used 32 male DS rats , 220 -230 g, 2-3 months old for this stude.

Line 241 22oC and 50 % humitity (omit comma, add and)

Line 241.  The rats had free access to water and...

Line 242.  Omit "mice" and substitute "rats".  

Line 243 one week after

Line 248.  It was then vortexed at 50 Hz

Line 249. two hours, storing at 4oC between injections and warming to 20oC.

Line 251.  but free to drink water

Line 252.  The caudal vein

Line 254.  with a constant

Line 256from the parietal lobe

Line 260 five minute intervals.

Line 272 toes, not fingers.  If you wish to specify front or back toes, then please do.

Line 272.  Muscle relaxation was judged

Line 274.  Hearing induction was evaluated by clapping hands near the ear

Line 276.  in five minute

Line 277. intervals

Line 290. Genomic DNA of brain cortex was obtained  using a DNA extraction kit.  The target fragment (omit and then)

Line 296 calculation of the

304.  Substitute blots for members

Line 313.  Paraffin sections were made of the cerebral cortex.  Be sure to not use the word executed.  This word is only used for prisoners.

Author Response

Response to Reviewer 1 Comments

Dear Reviewer, we have revised the manuscript according to your suggestion, and we have also sent the manuscript to the MDPI Author Services for English editing. All corrections have been marked in blue in the manuscript. Thank you very much for your support.

Reviewer 2 Report

In this reviewer's opinion, the manuscript requires significant editing, both in English language and style, as well as in overall messaging and data interpretation.  Emulsified isoflurane is not, indeed, a frequently used intravenous anesthetic.  While there are multiple papers, over many decades, exploring IV administration of various volatile anesthetic formulations, to this reviewer's knowledge its use has not become frequent, either in veterinary or human populations.

The title and overall conclusions of the manuscript overstate the implications of the data presented.  The authors do not show that the mechanism of anesthesia is the activation of GABAAR/STEP61 signaling pathway, and that this process was regulated by DNA methylation, merely that these pathways and resulting gene/protein expression levels change in the presence of the study drug.

The authors administered IV EISO, monitored anesthesia in the animals, and evaluated the mRNA and protein expression, in the cerebral cortex,  of NMDAR, GABA R, OPRM, ERK, p38, JNK, NFkB, STEP61, and Notch.  Their rtPCR, western blot data, methylation dat, and IF data, are clear and correctly interpreted.  NMDAR is lower, GABA is higher, OPRM is higher, etc.

The primary concern this reviewer has is the assignation of a causal relationship between these gene and protein expression levels and induction and maintenance of anesthesia.  While it is likely that this is the case, the authors have not done the experiments that would be required to fully support this claim.

The revisions that this author suggests do not necessarily require additional experiments, which is why they are classified as minor rather than major revisions.  However, it is suggested that the authors reassess the conclusions and messaging of the manuscript, such that they present the data that they have without overconcluding their results.  For example, perhaps the overall message that EISO increases GABA and decreases NMDA, with concomitant changes in promoter methylation.  More work is necessary to determine the order in which these downstream effectors change, as well as to identify causal relationships between these factors.  This paper provides compelling evidence to support these relationships, and lays excellent groundwork for continued experiments to address these and other questions involving the utility and activity of novel formulations of volatile anesthetics.

The manuscript will require significant editing for English language, grammar, and style, and the letters above the bars that denote statistical significance are a little confusing.

Author Response

Response to Reviewer 2 Comments

Thank you very much for your suggestions, and we have sent the manuscript to the MDPI Author Services for English editing. The corrections have been marked in blue in the revised manuscript.

Point 1: The relationship between induction anesthesia and maintenance of anesthetic dose.

Response 1:

In this study, the dose of EISO used for induction and maintaining anesthesia was the same. Group HD was 60 mg kg -1 min-1, group LD was 45 mg kg -1 min-1(line 270-273).

Point 2: Emulsified isoflurane is not, indeed, a frequently used intravenous anesthetic.

Response 2:

“frequent” is deleted in the revised manuscript (line 13).

Point 3: The title overstate the implications of the data presented.

Response 3:

The title of the manuscript has been revised (line 2).

Point 4: The overall conclusions overstate the implications of the data presented.

Response 4:

According to your suggestions, we have revised the conclusions in the manuscript in order to explain our experimental results (Line 240-245 and line 345-352).

Point 5: The assignation of a causal relationship between target genes and proteins expression levels.

Response 5:

It is not clear how EISO directly affects gene methylation rates from the literature available until now, we discuss more about the relationship between DNA methylation and protein expression in revised manuscript (line 219-224)

Point 6: The letters above the bars that denote statistical significance are a little confusing.

Response 6:

We revised the manuscript.